

# Seroprevalence and lethality by SARS-CoV-2 in indigenous populations of Latin America and the Caribbean: a systematic review

Akram Hernández-Vásquez[1], Fabian Chavez-Ecos[2], Antonio Barrenechea-Pulache[3], Daniel Comandé[4] and Guido Bendezu-Quispe[5]

[1] Centro de Excelencia en Investigaciones Económicas y Sociales en Salud, Vicerrectorado de Investigación, Universidad San Ignacio de Loyola, Lima, Peru
[2] Sociedad Científica de Estudiantes de Medicina de Ica, Universidad Nacional "San Luis Gonzaga", Ica, Peru
[3] Universidad Científica del Sur, Lima, Peru
[4] Institute for Clinical Effectiveness and Health Policy (IECS), Buenos Aires, Argentina
[5] Centro de Investigación Epidemiológica en Salud Global, Universidad Privada Norbert Wiener, Lima, Peru

Corresponding author
Antonio Barrenechea-Pulache, abarrenechea@cientifica.edu.pe

## ABSTRACT

**Background:** Latin America and the Caribbean (LAC) has presented some of the highest numbers of cases and deaths due to COVID-19 in the world. Even though indigenous communities represent 8% of the total population in this region, the impact of COVID-19 on this historically vulnerable population has only been briefly explored. Thus, this study aimed to estimate the seroprevalence and lethality attributable to SARS-CoV-2 in the indigenous population of LAC.

**Methods:** A systematic review was conducted utilizing multiple databases (registry PROSPERO: CRD42020207862). Studies published in English, Spanish or Portuguese were selected between December 1st, 2019, and April 14th, 2021. The evaluation of the quality of the study was carried out utilizing the Quality Assessment Tool for Observational Cohort and Cross-Sectional Studies. A qualitative synthesis of the data analyzed was conducted following the MOOSE and PRISMA declarations.

**Results:** Fifteen studies met the inclusion criteria. Eleven studies were carried out in a Brazilian population, three in a Mexican population, and one in a Colombian population. Four studies reported data about the seroprevalence of SARS-CoV-2 in indigenous populations of Brazil (range: 4.2–81.65%). Twelve studies reported lethality in indigenous people (eight in Brazil, three in Mexico, and one in Colombia). In Brazil, a lethality of 53.30% was described in a hospital setting and between 1.83% and 4.03% in community studies. In Mexico, the lethality of COVID-19 ranged between 16.5% and 19.9%. Meanwhile, in Colombia, a lethality of 3.41% was reported. Most studies were deemed to be of good quality.

**Conclusions:** Despite COVID-19 affecting indigenous populations of LAC, there is limited evidence of the seroprevalence and lethality of the infection by SARS-CoV-2 in this population. Future investigations should ensure standardized methods that allow comparability among studies and ensure the precision of the results obtained.

# INTRODUCTION

Since the detection of the first case of SARS-CoV-2 in Wuhan, China, in 2019, the propagation of this virus has increased until being declared a pandemic on the 11th of March 2020 (*World Health Organization, 2019*). The most common mode of transmission is through respiratory droplets and prognosis of the patient depends on several factors including age, severity of disease presentation, comorbidities and response to treatment (*Zhou et al., 2021*). One of the groups most vulnerable to SARS-CoV-2 is the indigenous population due to the presence of social determinants at an individual, community, and societal level that condition lower health quality. These include higher levels of poverty, malnutrition, lower levels of education, poor access to essential sanitation services, difficulties accessing health care services due to geographical and cultural barriers, and low availability of adequately equipped health care establishments (*McLeod et al., 2020*; *Charlier & Varison, 2020*; *United Nations, 2021*). Likewise, immunological factors and chronic diseases have been reported to increase the susceptibility to infectious disease outbreaks in this group of people (*Gelaye et al., 2020*). A negative effect on the health status of indigenous people has been described in recent decades due to these diseases with lethality up to six times that found in the general population (*McLeod et al., 2020*; *La Ruche et al., 2009*; *Food and Agriculture Organization, 2020*).

Latin America and the Caribbean (LAC) has presented some of the highest numbers of cases and deaths due to COVID-19 in the world (*Johns Hopkins University, 2021*). The first regional case of SARS-CoV-2 infection was reported in February 2020 in Brazil, while the first death due to COVID-19 was reported in Argentina in March (*Rodriguez-Morales et al., 2020*). In LAC, there are over 800 indigenous communities, and together they account for approximately 60 million people, representing 8% of the total population. Thus, LAC presents the highest proportion of indigenous people compared to other inhabitants (*United Nations Office for the Coordination of Humanitarian Affairs, 2020*). Indigenous populations have conditions that make them vulnerable to the COVID-19 pandemic. These conditions include high levels of poverty, which affect 43% of indigenous households, an average salary 31% lower than that of other workers and lower levels of education with reports indicating that only 20% of indigenous populations obtain conventional education and lower access to health care (*United Nations, 2021*; *United Nations Office for the Coordination of Humanitarian Affairs, 2020*). It was to be expected that these historical inequalities would increase during the COVID-19 pandemic, causing indigenous people to be left behind when distributing scarce resources, such as diagnostic tests, personal protective equipment, mechanical ventilators, and medication necessary for the treatment of critically ill patients (*United Nations, 2021*; *Food and Agriculture Organization, 2020*), being a scenario that increases the risk of severe disease and death in this group. Thus, tending to the needs of indigenous populations in LAC continues to be a challenge and has been made even greater due to the current events of the pandemic.

To date, an ever-growing number of COVID-19 cases has been reported in all LAC countries (*Johns Hopkins University, 2021*), including cases among indigenous people in this region (*United Nations Office for the Coordination of Humanitarian Affairs, 2020*; *Pan American Health Organization, 2021a*). Regardless, the presence and the impact of the COVID-19 pandemic on mortality in indigenous populations of LAC have seldom been studied. Thus, we carried out a systematic review to estimate the seroprevalence and lethality attributable to COVID-19 in indigenous populations of (LAC). This information could serve as evidence to generate strategies oriented towards better allocation of health resources in this population.

# METHODS

This systematic review was registered in the PROSPERO database (registration code: CRD42020207862) and reported using the guidelines of the Meta-analysis of Observational Studies in Epidemiology (MOOSE) (*Stroup et al., 2000*) and Preferred Reporting Items for Systematic Reviews and Meta-Analyses (PRISMA) statements (*PRISMA, 2015*).

## Study inclusion and exclusion criteria

We used the CoCoPop approach (Condition, Context, and Population) to establish inclusion and exclusion criteria in studies describing seroprevalence and lethality due to SARS-CoV-2 (*Munn et al., 2015*). All observational studies published in English, Spanish or Portuguese between December 1st, 2019, and April 14th, 2021, that estimated the prevalence and lethality of COVID-19 among indigenous populations of LAC regardless of age, were eligible for inclusion. All observational study designs that estimate prevalence and lethality in this population were selected for review. This time frame was chosen because the first group of COVID-19 cases in China was reported in December 2019 (*Burki, 2020*). Publications such as systematic reviews, review articles, congressional acts, letters, commentaries, and editorials were excluded. Gray literature was not included (the search was restricted to publications evaluated by a peer review process to maintain rigor and ensure the quality of the selected studies).

## Data sources

Searches were conducted in the following databases: PubMed, Embase (Excerpta Medica Database), CINAHL (Cumulative Index to Nursing and Allied Health Literature), Web of Science, Scopus, Google Scholar, and the regional database *LILACS (Literatura Latinoamericana y del Caribe en Ciencias de la Salud)* to identify studies that were of interest. While utilizing Google Scholar, the number of results reviewed was limited to the first 10 pages following the recommendations on using this database to conduct a systematic review (*Haddaway et al., 2015*; *Bramer et al., 2017*).

## Study Outcomes: Lethality and seroprevalence

For our review, lethality was defined as the proportion of indigenous patients diagnosed or suspected of having COVID-19 who died during the study period as a result of said disease; this was often reported as a percentage among the findings of the cited article.

Likewise, seroprevalence was defined as the proportion of individuals tested for and found to possess COVID-19 antibodies; this was also often reported in the cited article.

## Search strategies and screening

The research team developed the search strategies with the guidance of a librarian with experience in medical investigation (DC). These strategies were reviewed and approved by all researchers. After approval of the search strategies, one author (DC) executed the search, compiled the results and eliminated duplicates. Date and publication status were not considered as restrictions. The search was limited to studies published in English, Spanish or Portuguese since these are the official languages of LAC countries and are widely utilized to inform studies in this region. Furthermore, searches were conducted using the reference lists of all relevant articles. The search strategy is described in Appendix S1.

The search results were exported to EndNote X9 (Thompson and Reuters, Philadelphia, USA) to eliminate duplicate publications. Next, titles and/or abstracts of the retrieved publications were screened independently by two review authors (AHV and ABP) to identify studies that potentially met the inclusion criteria of this review using the web application Rayyan (https://rayyan.qcri.org/). Each selected publication was classified into the following categories: excluded, maybe, and included. In the case of maybe or a conflict about the inclusion of any publication, the reviewers discussed and achieved consensus regarding inclusion or exclusion of that publication.

All studies included by titles and abstracts entered the full-text evaluation phase. These studies were evaluated independently by the same members of the review team. In the case of conflict between reviewers, this was discussed, and consensus was achieved regarding the inclusion or exclusion of that publication.

## Data extraction

Two review authors (AHV and ABP) extracted the relevant data independently using a standardized data extraction sheet. In the case of disagreement, this was resolved by a discussion between the two authors. The following data were extracted from each study: date of publication, first author, journal name, type o publication, language, study objectives, period of data collection, country, size and sample characteristics, operational definition of a COVID-19 case, and data on seroprevalence and lethality in COVID-19 patients.

## Quality assessment

Two reviewers (AHV and ABP) independently evaluated the quality of the studies included. In the case of disagreement, this was resolved by discussion between the two authors. The quality assessment was conducted utilizing the Quality Assessment Tool for Observational Cohort and Cross-Sectional Studies of the National Institutes of Health (*National Heart, Lung, and Blood Institute, 2021*). This tool considers 14 criteria to evaluate the risk of bias in selection, information, measure, and confusion. Each of the studies included was evaluated with these 14 criteria ("Yes" was considered in the case that

it complied with the criteria; "No", in the case that it did not comply with the criteria; "NA", when the criteria did not apply to the study; "ND", when the criteria were not possible to determine; and "NR", when the information regarding the evaluated criteria was not reported). Following the methodology for objective evaluation of the quality employed in previous systematic reviews that have utilized this tool, each criterion assigned "Yes" summed one point to the overall score and criteria evaluated, while no points were added for "No" and "ND" (0 points). The criteria qualified as "NA" did not count towards the percentage of points of the maximum total possible. The quality of a study was evaluated according to the percentage of the maximum score obtained (>50% good, 30–50% regular, <30% bad). (*Ahmed et al., 2020*; *Hinds et al., 2019*).

### Data synthesis

We utilized a narrative synthesis focus. Findings were presented in summary tables, following the MOOSE guideline (*PRISMA, 2015*). Descriptive tables were constructed with information on the seroprevalence and/or lethality of SARS-CoV-2 in groups of the indigenous population of LAC. Descriptive summaries of the results for indigenous populations, including country, sex, age, comorbidities, and ethnic group, were reported.

### Ethics

Ethics approval was not required for this study because it was based on published studies.

### Funding

This study obtained a grant from the *Universidad Privada Norbert Wiener*. The funder did not have any role in the study design, data collection, data synthesis, analysis, or preparation of the manuscript.

## RESULTS

### Search results

We obtained a total of 747 unique results from the search after eliminating duplicates. After revision by title and abstract, 26 full-text articles were read. Following evaluation, one study was found to be an editorial, and 10 did not provide data on indigenous people, and were, thus, excluded. In the end, 15 studies complied with the inclusion criteria and are reported below (Fig. 1).

### Study characteristics

All the studies included were published between 2020–2021 and were cross-sectional ($n = 15$). Eleven of the included studies were conducted in the indigenous population of Brazil (*Hallal et al., 2020*; *Baqui et al., 2020*; *Santos et al., 2020*; *Palamim, Ortega & Marson, 2020*; *Horta et al., 2020*; *Rodrigues et al., 2021*; *da Silva et al., 2021a*; *Escobar, Rodriguez & Monteiro, 2020*; *Mendes et al., 2021*; *da Silva et al., 2021b*; *Hillesheim et al., 2020*), three were conducted in the indigenous population of Mexico (*Ortiz-Hernández & Pérez-Sastré, 2020*; *Argoty-Pantoja et al., 2021*; *Ibarra-Nava et al., 2021*), and one in Colombia (*Cifuentes et al., 2021*). No multicounty studies were included. The studies that reported lethality included sample sizes of the indigenous people infected with

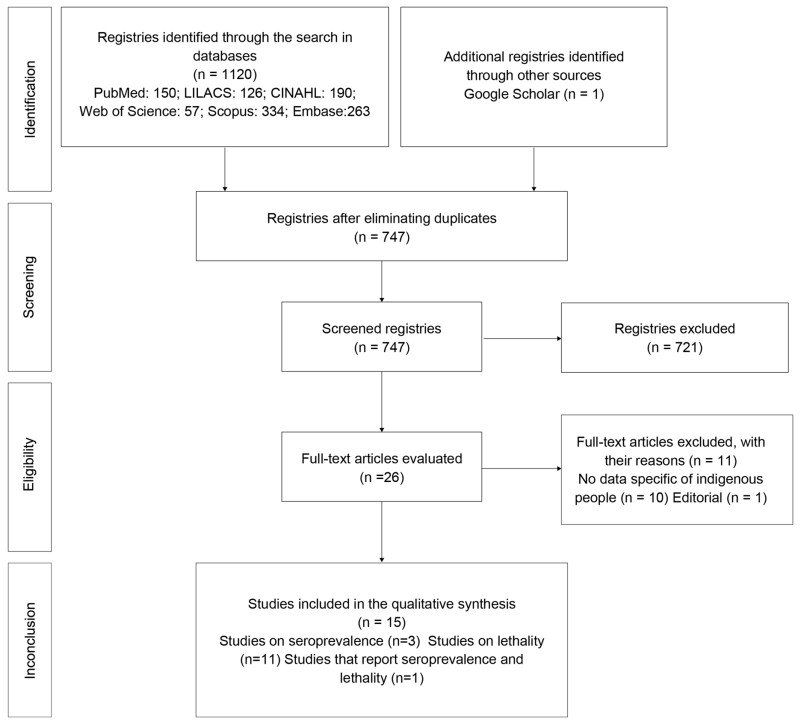

**Figure 1 Flowchart of the selection of studies according to the PRISMA guidelines.**

COVID-19 which varied from 15 to 41,855 individuals, while those that reported seroprevalence tested samples of indigenous people that ranged from 100 to 2,890 individuals. Likewise, most studies were published in English; only one was published in Portuguese (*da Silva et al., 2021a*), and two were also available in Spanish (*Horta et al., 2020*; *Ortiz-Hernández & Pérez-Sastré, 2020*). Only five of the included studies specified data about the age and sex of the indigenous participants studied (*Rodrigues et al., 2021*; *da Silva et al., 2021a*; *da Silva et al., 2021b*; *Argoty-Pantoja et al., 2021*; *Ibarra-Nava et al., 2021*). Six of the studies were conducted specifically in the indigenous population, all from Brazil (*Santos et al., 2020*; *Palamim, Ortega & Marson, 2020*; *Rodrigues et al., 2021*; *da Silva et al., 2021a*; *Mendes et al., 2021*; *da Silva et al., 2021b*). Three pairs of studies analyzed the same database during a similar time frame; thus, the results of one study may have been included in those of another (*Hallal et al., 2020*; *Baqui et al., 2020*; *Horta et al., 2020*; *da Silva et al., 2021a*; *Argoty-Pantoja et al., 2021*; *Ibarra-Nava et al., 2021*) (Table 1).

## Results related to the seroprevalence of SARS-CoV-2 infection

Four studies (*Hallal et al., 2020*; *Horta et al., 2020*; *Rodrigues et al., 2021*; *da Silva et al., 2021b*) reported data about the seroprevalence of SARS-CoV-2 in indigenous populations in Brazil (Table 1). The two largest studies conducted household surveys of 1,219 indigenous people through three surveys and 767 indigenous people in two surveys, respectively. *Horta et al. (2020)* reported a seroprevalence of 5.4% for the three surveys, while *Hallal et al. (2020)* reported a seroprevalence of 4.2% (95% CI [2.2–7.2]) in the first survey and 6.3% (95% CI [4.2–9.2]) in the second survey, after adjusting for the design and

**Table 1 Extraction of data on seroprevalence and lethality of SARS-CoV-2 among indigenous people from the studies selected.**

| Study | Country | Study design | Data source | Seroprevalence | | | | | Lethality | | | | |
|---|---|---|---|---|---|---|---|---|---|---|---|---|---|
| | | | | Setting | Test | Sample | Mean age (SD)/ Sex (%) | Prevalence (N of individuals with positive antibody test/sample) | Setting | Test | COVID-19 Cases | Mean age (SD)/Sex (%) | Lethality (N of COVID-19 patients diseased/N of COVID-19 Cases) |
| (1) *Baqui et al. (2020)* | Brazil | Cross sectional | SIVEP-Gripe (public dataset from Feb 27 to May 4, 2020) | ND | ND | ND | ND | ND | Hospital | RT-PCR | 15 | ND | 8/15 = 53.33% |
| (2) *Hallal et al. (2020)* | Brazil | Cross sectional | Household survey (conducted in 2 dates: May 14–21, and June 4–7, 2020.) | Community | WONDFO SARS-CoV-2 Antibody Test | Survey 1: 327 Survey 2: 440 | ND | Survey 1: 12/327 = 4.2%* 95% CI [2.2–7.2] Survey 2: 24/440 = 6.3%* 95% CI [4.2–9.2] | ND | ND | ND | ND | ND |
| (3) *Ortiz-Hernández & Pérez-Sastré (2020)* | Mexico | Cross sectional | Data obtained by the *Dirección General de Epidemiología de la Secretaría de Salud del Gobierno Federal* of Mexico (data up to July 10 2020) | ND | ND | ND | ND | ND | Hospital | Not specified: confirmed and suspicious cases were included | 1% (total numbers not specified) | ND | 19.9% (total numbers not specified) |
| (4) *Palamim, Ortega & Marson (2020)* | Brazil | Cross sectional | Special Secretariat for Indigenous Health (SESAI) database (data up to June 5 2020) | ND | ND | ND | ND | ND | National | Not specified: confirmed and suspicious cases were included | 2,157 (confirmed cases = 1,737; suspected cases = 420) | ND | 70/2,157= 3.25% |
| (5) *Santos et al. (2020)* | Brazil | Cross sectional | Microdata catalog and official bulletins for each Brazilian state between 26 February and 28 August 2020 | ND | ND | ND | ND | ND | Community | Not specified: confirmed and suspicious cases were included | 29,008 | ND | 532/29,008= 1.83% |
| (6) *Horta et al. (2020)* | Brazil | Cross sectional | Household survey (conducted from May 14-21, June 4-7 and June 21-24) | Community | WONDFO SARS-CoV-2 Antibody Test | 1,219 | ND | 66/1,219 = 5.4% | ND | ND | ND | ND | ND |

(Continued)

| Study | Country | Study design | Data source | Seroprevalence | | | | | Lethality | | | | |
|---|---|---|---|---|---|---|---|---|---|---|---|---|---|
| | | | | Setting | Test | Sample | Mean age (SD)/ Sex (%) | Prevalence (N of individuals with positive antibody test/sample) | Setting | Test | COVID-19 Cases | Mean age (SD)/Sex (%) | Lethality (N of COVID-19 patients diseased/N of COVID-19 Cases) |
| (7) Argoty-Pantoja et al. (2021) | Mexico | Longitudinal | SISVER (February 27, 2020 until July 30, 2020) | ND | ND | ND | ND | ND | National | SARS-CoV-2 infection certified by the Institute of Epidemiological Diagnosis and Reference (InDRE) | 4,469 | Age 50.4 (17.4) Male 2,656 (59.4%) | 768/ 4,469=17.2% |
| (8) Cifuentes et al. (2021) | Colombia | Retrospective cohort | SIVIGILA (2 March 2020 to 26 October 2020) | ND | ND | ND | ND | ND | National | RT-PCR | 22,787 | ND | 776/22,787 = 3.41% |
| (9) da Silva et al. (2021a) | Brazil | Cross sectional | SIVEP-Gripe (January 1st to June 16 2017, 2018, 2019 and 2020) | ND | ND | ND | ND | ND | National | ND | 318 | Male 195 (61.32%) | 155/318 = 48.74% |
| (10) Escobar, Rodriguez & Monteiro (2020) | Brazil | Cross sectional | Brazilian National Health System Epidemiological Surveillance System Computerization Strategy (January 1 and August 20, 2020) | ND | ND | ND | ND | ND | National | ND | 843 | ND | 12/843 =1.42% |
| (11) Hillesheim et al. (2020) | Brazil | Cross sectional | SIVEP-Gripe (Epidemiological Weeks 1 to 38, 2020 (up to 19/ 09/2020)) | ND | ND | ND | ND | ND | National of hospitalized patients | Laboratory diagnosis – not specified | 113 | Study included children and adolescents (not specified) | 26/113 = 23% |
| (12) Ibarra-Nava et al. (2021) | Mexico | Cross sectional | SISVER (February 28 to August 3, 2020) | ND | ND | ND | ND | ND | National | RT-PCR | 4,178 | Non survivors: Age: 63.4 (13.0) Male 461 (18.5%) | 691/4,178 = 16.5% |

| Study | Country | Study design | Data source | Seroprevalence | | | | | Lethality | | | | |
|---|---|---|---|---|---|---|---|---|---|---|---|---|---|
| | | | | Setting | Test | Sample | Mean age (SD)/Sex (%) | Prevalence (N of individuals with positive antibody test/sample) | Setting | Test | COVID-19 Cases | Mean age (SD)/Sex (%) | Lethality (N of COVID-19 patients diseased/N of COVID-19 Cases) |
| (13) Mendes et al. (2021) | Brazil | Cross sectional | Press releases by the Health Ministry of Brazil | ND | ND | ND | ND | ND | National | Laboratory analysis (not specified) and by clinical criteria | 41,855 | ND | 549/41,855 = 1.31% |
| (14) Rodrigues et al. (2021) | Brazil | Cross sectional | Survey conducted on Xikrin of Bacajá people | Community | Rapid test (lateral flow method; Guangzhou Wondfo Biotech Co., China) and an enzyme-linked immunosorbent assay (ELISA; Anti-SARS-CoV-2 S1 IgG, Euroimmun, Brazil) | 100 | Age 27.9 (range 8–82) Men 51 | 58 were IgG-reactive (58%) by a rapid test, and 73 (73%) were reactive in an enzyme-linked immunosorbent assay | ND | ND | ND | ND | ND |
| (15) da Silva et al. (2021a) | Brazil | Cross sectional | Communities in the municipality of Amaturá, Amazonas (January and August 2020) | Community | Rapid test (not specified) | 2,890 inhabitants | ND | Of the total inhabitants 109 were suspected of being infected with SARS-CoV-2, of which 89 were confirmed = 81.65% | Community | Rapid test (not specified) | 89 | Age 32.4 (±23.6) Female 50 (56.2%) | 0/89 |

**Notes:**

ND, No Data; SIVEP-Gripe, Influenza Epidemiological Surveillance Information System; SISVER, Epidemiological Surveillance System for Viral Respiratory Diseases; RT-PCR, Reverse transcriptase polymerase chain reaction; SIVIGILA, National Public Health Surveillance System.
* Adjusted for the design and validity of the test.

validity of the diagnostic test. The third study conducted surveys in the Xikrin of Bacajá indigenous population (Kayapó) and employed both the WONDFO and ELISA Anti-SARS-CoV-2 S1 IgG test in a sample of 100 indigenous individuals, 58% of whom were IgG reactive by a rapid test, and 73% were reactive in an enzyme-linked immunosorbent assay (*Rodrigues et al., 2021*). Finally, the fourth study was conducted in various indigenous communities in Amaturá, Amazonas. In this study, a total of 109 individuals with respiratory symptoms were tested using an unspecified rapid test, 89 of whom tested positive with a seroprevalence of 81.65% (*da Silva et al., 2021b*).

## Results related to lethality by SARS-CoV-2

Twelve studies reported data on the lethality in indigenous populations; eight were from Brazil (*Baqui et al., 2020*; *Santos et al., 2020*; *Palamim, Ortega & Marson, 2020*; *da Silva et al., 2021a*; *Escobar, Rodriguez & Monteiro, 2020*; *Mendes et al., 2021*; *da Silva et al., 2021b*; *Hillesheim et al., 2020*), three from Mexico (*Ortiz-Hernández & Pérez-Sastré, 2020*; *Argoty-Pantoja et al., 2021*; *Ibarra-Nava et al., 2021*), and one from Colombia (*Cifuentes et al., 2021*) (Table 1). Regarding the Brazilian studies, lethality ranged between 0%, in a study that reported zero deaths among 89 confirmed COVID-19 cases (*da Silva et al., 2021a*), and 53.3% in a study that included 15 indigenous people with COVID-19 in a hospital setting, eight of whom died (*Baqui et al., 2020*). *Mendes et al. (2021)* conducted a cross-sectional study including the largest sample of COVID-19 cases among indigenous individuals ($n = 41,855$), among whom 549 died, reporting a lethality of 1.31%. In Mexico, *Ibarra-Nava et al. (2021)* reported that 691 people of a total of 4,178 COVID-19 indigenous cases died, obtaining a lethality of 16.5%. *Argoty-Pantoja et al. (2021)* found that 768 out of 4,469 infected indigenous people died, reporting a lethality of 17.2%, and *Ortiz-Hernández & Pérez-Sastré (2020)* reported lethality of 19.9% (though this publication did not specify the exact number of deaths). Lastly, in the study conducted in Colombia, it was reported that 776 of a total of 22,787 infected indigenous individuals died, with lethality of 3.4% (*Cifuentes et al., 2021*).

## Study quality

Most studies were qualified as having good quality (>50% of the maximum score utilizing the Quality Assessment Tool for Observational Cohort and Cross-Sectional Studies of the National Institutes of Health). Only three were found to have regular quality (50%) (Table 2).

## DISCUSSION

This systematic review aimed to summarize the evidence available on the seroprevalence and lethality of infection by SARS-CoV-2 in the indigenous populations of LAC. Only a few non-representative studies measuring outcomes of COVID-19 in the indigenous people of LAC countries were available. In general, only three Latin American countries (Brazil, Mexico, and Colombia) have studied and reported data on seroprevalence and lethality in indigenous populations, including studies at community (12/15) and hospital levels (3/15). Likewise, only 12 studies were designed to evaluate the lethality of COVID-19

**Table 2 Evaluation of the quality of the studies included ($n$ = 15).**

| Study | P1 | P2 | P3 | P4 | P5 | P6 | P7 | P8 | P9 | P10 | P11 | P12 | P13 | P14 | Rating |
|---|---|---|---|---|---|---|---|---|---|---|---|---|---|---|---|
| *Baqui et al. (2020)* | Yes | Yes | Yes | Yes | No | No | No | NA | Yes | NA | Yes | NA | NA | Yes | 70.0% |
| *Hallal et al. (2020)* | Yes | Yes | Yes | Yes | No | No | No | NA | Yes | NA | Yes | No | NA | Yes | 63.6% |
| *Ortiz-Hernández & Pérez-Sastré (2020)* | Yes | Yes | Yes | Yes | No | No | No | NA | Yes | NA | No | NA | NA | Yes | 60.0% |
| *Palamim, Ortega & Marson (2020)* | Yes | Yes | Yes | Yes | No | No | No | NA | Yes | NA | No | NA | NA | NA | 62.5% |
| *Santos et al. (2020)* | Yes | Yes | Yes | No | NA | No | No | NA | NA | NA | No | NA | NA | NA | 50.0% |
| *Horta et al. (2020)* | Yes | Yes | Yes | Yes | Yes | No | No | NA | Yes | NA | Yes | No | NA | Yes | 72.7% |
| *Argoty-Pantoja et al. (2021)* | Yes | Yes | Yes | Yes | No | No | No | NA | Yes | NA | Yes | NA | NA | Yes | 70.0% |
| *Cifuentes et al. (2021)* | Yes | Yes | Yes | Yes | No | No | No | NA | Yes | NA | Yes | NA | NA | No | 60.0% |
| *da Silva et al. (2021a)* | Yes | Yes | Yes | Yes | No | No | No | NA | Yes | NA | Yes | NA | NA | NA | 66.7% |
| *Escobar, Rodriguez & Monteiro (2020)* | Yes | Yes | Yes | Yes | No | No | No | NA | Yes | NA | Yes | NA | NA | No | 60.0% |
| *Hillesheim et al. (2020)* | Yes | Yes | Yes | Yes | No | No | No | NA | Yes | NA | Yes | NA | NA | No | 60.0% |
| *Ibarra-Nava et al. (2021)* | Yes | Yes | Yes | Yes | No | No | No | NA | Yes | NA | Yes | NA | NA | Yes | 70.0% |
| *Mendes et al. (2021)* | Yes | Yes | Yes | No | NA | No | ND | NA | NA | NA | Yes | NA | NA | NA | 57.1% |
| *Rodrigues et al. (2021)* | Yes | Yes | ND | Yes | No | No | No | NA | Yes | NA | Yes | No | NA | NA | 50.0% |
| *da Silva et al. (2021a)* | Yes | Yes | No | Yes | No | No | No | NA | Yes | NA | Yes | No | NA | NA | 50.0% |

**Note:**
NA, Non-applicable; ND, not possible to determine.

in the indigenous population (eight in Brazil). In general, the studies found had a cross-sectional design. The seroprevalence of SARS-CoV-2 was only reported in the indigenous population of Brazil, with values between 4.2 and 81.65%. Regarding lethality, higher values were reported in studies carried out in hospital settings (19.9% to 53.33%) compared to those conducted in a community setting (0% to 48.74%).

Regarding the seroprevalence of indigenous populations, the studies identified only evaluated the Brazilian indigenous population using small sample sizes; the study with the largest sample reported a seroprevalence of 5.4%. After reviewing the available literature, we found that there is also scarce information regarding the seroprevalence of SARS-CoV-2 in the indigenous populations of other regions of the world. One study in the indigenous population of Alaska reported that the prevalence of SARS-CoV-2 infection reached 1.3% (*Hatcher et al., 2020*), a value within the range found in studies in the indigenous population of Brazil reported in this review. We emphasize that Brazil is one of the countries with the highest number of cases of COVID-19 in the world (3rd place), with approximately 19 million cases by July 2021 (*Johns Hopkins University, 2021*). Studies presenting results regarding the seroprevalence of SARS-CoV-2 in other LAC countries were not found. However, in countries such as Bolivia (*Fondo para el Desarrollo de los Pueblos Indígenas de América Latina y el Caribe, 2020a*), Peru (*Centro Nacional de Epidemiología Prevención y Control de Enfermedades, 2021*), Ecuador (*Castro, 2020*), Venezuela (*Pan American Health Organization, 2021b*), and Colombia (*Organización Panamericana de la Salud, 2020*), cases of COVID-19 have been reported in the indigenous populations. During the COVID-19 pandemic, a deficit of diagnostic tests, lack of masks,

and the suspension of health programs in indigenous communities have been described (*Reinders et al., 2020*), being a situation that could lead to a decline in health care in these communities as well as an underestimation of the prevalence of COVID-19 in indigenous populations. Thus, health care for this historically vulnerable group should be prioritized during times of crisis. Additionally, given that most countries report official information on COVID-19 without disaggregating it according to ethnicity (*Fondo para el Desarrollo de los Pueblos Indígenas de América Latina y el Caribe, 2020b*), it is necessary to overcome the deficiencies in the data recording process to evaluate the impact of COVID-19 or other diseases in indigenous communities.

Regarding lethality, differences were found among the studies evaluating the Brazilian population (23% and 53.3% in the hospital setting; 0% to 48.74% in the community setting) and the Mexican population (19.9% in the hospital setting). The difference in the lethality among the studies reported and the fact that higher lethality has been reported in hospital settings could be due to indigenous patients with more severe forms of COVID-19 being concentrated in the hospital setting, raising the lethality due to COVID-19 among the patients that received medical care in a hospital setting. Until July 13th, 2021, Brazil (534,233) and Mexico (235,058) reported two of the highest numbers of deaths due to COVID-19 worldwide (second and fourth place, respectively) (*Johns Hopkins University, 2021*). Regarding the number of COVID-19 cases in these countries, the lethality due to COVID-19 in the general population in Mexico is the highest in the world with 9.06%, being 2.80% in Brazil (*Johns Hopkins University, 2021*). Similar to the outcomes of seroprevalence of SARS-CoV-2, we found no study on lethality due to COVID-19 in other countries of LAC. Nevertheless, deaths due to COVID-19 in the indigenous population of other Latin American countries such as Bolivia, Peru, Colombia, and Ecuador show hundreds of deaths due to COVID-19 in indigenous communities (*Fondo para el Desarrollo de los Pueblos Indígenas de América Latina y el Caribe, 2020a*; *Castro, 2020*; *Organización Panamericana de la Salud, 2020*; *Fondo para el Desarrollo de los Pueblos Indígenas de América Latina y el Caribe, 2021*). The trust for the development of indigenous communities in LAC stated that by December 2020 over 73,000 cases of COVID-19 and 21,000 deaths had been reported among indigenous communities of the amazon region. Many of these communities were already vulnerable due to poor levels of sanitation and dependence on income from daily manual labor, and many were left to fend for themselves due to being geographically isolated opting to turn towards traditional medicine as a means of combatting illness (*Fondo para el Desarrollo de los Pueblos Indígenas de América Latina y el Caribe, 2021*). Considering the potential sub-registration of cases and deaths due to COVID-19 in these countries, the predisposition of indigenous populations, and limitations in the availability of diagnostic tests, COVID-19 may be an important cause of mortality in indigenous people.

To date, international organizations such as the Pan American Health Organization have required governments of LAC countries to implement measures to slow the spread of COVID-19 in indigenous communities, including increasing the number of health workers and medical supplies, as well as providing diagnostic tests for COVID-19 and vaccines and treatments for this disease when available (*Pan American Health*

*Organization, 2021a*). The need to adopt emergency measures to protect this population was demonstrated in Brazil, where a supreme court (August 2020) exposed the Brazilian government's failure to protect the indigenous population from the impact of COVID-19 after an indigenous leader had died from this disease (*Álvares, 2021*). It should be noted that in countries with indigenous communities in the Amazon area, including Brazil, Colombia, and Peru, the presence of illegal extractive activities generates a great environmental and direct impact on the health of these communities by exploiting the resources used for their subsistence and requiring them to move from their territory, likely facilitating the introduction and spread of COVID-19 (*Villén-Pérez et al., 2020*; *Langlois, 2020*; *Swenson et al., 2011*). Regarding the previous issue, the Brazilian court also pointed out the need to expel illegal gold miners from protected areas (*Álvares, 2021*). Therefore, it is urgent that governments take measures to control activities that could introduce COVID-19 and cause environmental impacts that affect the health of indigenous communities.

Indigenous communities have taken an active role in their protection against COVID-19. In LAC, for example, it is described that indigenous communities in Brazil, such as Paiter Suruí and Parque Ingidena do Xingu established voluntary quarantines in March 2020 (*Charlier & Varison, 2020*). Furthermore, in Peru, it has been described that the majority of the communities of the rural districts Nauta, Parinari, and Sequena prohibited the entry of people not belonging to the community. However, half of these communities did allow residents to travel to the cities for commercial and social reasons, and the prohibition of social events was only abided by a third of the communities since the control measure, decreed by the Peruvian government, competes with their daily routine (*Reinders et al., 2020*). Although it is described that the measures taken by governments to control the spread of COVID-19 towards indigenous communities have been deficient (*Fondo para el Desarrollo de los Pueblos Indígenas de América Latina y el Caribe, 2020b*), it should be emphasized that mistrust in authority and disbelief in the existence and severity of the pandemic likely influenced outcomes (*Fondo para el Desarrollo de los Pueblos Indígenas de América Latina y el Caribe, 2020b*; *Meneses-Navarro et al., 2020*). Therefore, effective, culturally appropriate communication strategies with the communities should be established to transmit accurate information about COVID-19 (*Meneses-Navarro et al., 2020*). These strategies should also include the participation of community representatives (*United Nations Office for the Coordination of Humanitarian Affairs, 2020*; *Fondo para el Desarrollo de los Pueblos Indígenas de América Latina y el Caribe, 2020b*). Authorities could also establish differentiated lockdowns according to the number of individuals affected by COVID-19 in the indigenous communities to reduce the impact in the activities of these populations.

This review has some limitations. Although we sought to be comprehensive in terms of information sources, including a regional bibliographic database such as LILACS as well as a search engine such as Google Scholar and including documents in the main languages of the LAC countries, there may be publications in the gray literature on the study subject that would not have been collected as they were not indexed in the information sources used. Likewise, we found few studies from few LAC countries

examining COVID-19 in the indigenous population. Also, there were limitations regarding the information provided in the studies evaluated; for example, in some cases, no information on the recruitment criteria of the participants or type of test used to detect positive SARS-CoV-2 cases was available. Furthermore, there is variability in the reporting of results, which did not allow a meta-analysis to be carried out. Despite this, being that this study is the first systematic review to summarize the evidence available on the seroprevalence and lethality of SARS-CoV-2 infection in the indigenous population of LAC, we consider that the use of a search strategy with exhaustive key terms allowed an adequate approach to the problem under study.

## CONCLUSIONS

In conclusion, this systematic review found that there is limited evidence on the seroprevalence and lethality of SARS-CoV-2 in the indigenous population of LAC. The few studies identified described the presence of COVID-19 in indigenous people of Brazil, Mexico, and Colombia. COVID-19 is a cause of death in these LAC indigenous populations, although there were differences in the lethality reported in hospital and community settings during the pandemic. In order not to leave anyone behind, future investigations with a prospective design should aim to use standardized methods that allow comparability among the studies of seroprevalence and lethality of SARS-CoV-2 in this population, this data should be made readily available to allow evaluation of the true impact of COVID-19 in indigenous people, who have historically had inequality in access to medical care.

### Funding
This study obtained a grant from the Universidad Privada Norbert Wiener (No. 118-2021-R-UPNW). The funders had no role in study design, data collection and analysis, decision to publish, or preparation of the manuscript.

### Grant Disclosures
The following grant information was disclosed by the authors:
Universidad Privada Norbert Wiener: 118-2021-R-UPNW.

### Competing Interests
The authors declare that they have no competing interests.

### Author Contributions
- Akram Hernández-Vásquez conceived and designed the experiments, performed the experiments, analyzed the data, prepared figures and/or tables, authored or reviewed drafts of the paper, and approved the final draft.
- Fabian Chavez-Ecos conceived and designed the experiments, performed the experiments, prepared figures and/or tables, authored or reviewed drafts of the paper, and approved the final draft.

- Antonio Barrenechea-Pulache performed the experiments, analyzed the data, prepared figures and/or tables, authored or reviewed drafts of the paper, and approved the final draft.
- Daniel Comandé conceived and designed the experiments, performed the experiments, analyzed the data, prepared figures and/or tables, authored or reviewed drafts of the paper, executed the bibliographic search and eliminated duplicate registries, and approved the final draft.
- Guido Bendezu-Quispe conceived and designed the experiments, performed the experiments, prepared figures and/or tables, authored or reviewed drafts of the paper, and approved the final draft.

### Data Availability

The search strategies are available in Appendix S1.

### Supplemental Information

Supplemental information for this article can be found online at http://dx.doi.org/10.7717/peerj.12552#supplemental-information.

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
