# Peer review of "Seroprevalence and lethality by SARS-CoV-2 in indigenous populations of Latin America and the Caribbean: a systematic review"

_PeerJ, doi:10.7717/peerj.12552_

## Round 0.1 · original submission · Minor Revisions

This study addresses a very important and timely topic as it relates to the ongoing pandemic. Though the reviewers raised several issues regarding the presentation of the manuscript and the analyses, it is my judgment that these are minors issues that could be easily addressed by the authors. In addition to providing a point-by-point response to reviewers' comments, the readability of the manuscript will be greatly improved by additional English proofreading.

Reviewer 1 ·

Basic reporting

no comment

Experimental design

no comment

Validity of the findings

no comment

Additional comments

Comments:
In a manuscript titled “Seroprevalence and lethality by SARS-CoV-2 in indigenous populations of Latin America and the Caribbean: A systematic review (#63932)”; since that COVID-19 is currently a global health threat and an international public health emergency, knowledge about seroprevalence and mortality of current emerging disease in indigenous and populous communities is important . Thus, topic is interesting. There are some points that I mentioned for author(s);
Abstract :
1- In background section explained just about aim of study. Recommended that change the title of this section to objectives( if it is accordance with instruction of manuscript preparation of journal) or to complete using an additional line about background of your subject.

Method :

1- What was your strategy for facing with grey literatures, such as government reports and non-peer reviewed manuscript? Did you include non-peer reviewed studies in your review? Please explain in your manuscript.

2- What was your inclusion or exclusion criteria for selecting the under study population of articles? In other words, you reviewed the articles that just conducted on adolescents? Or general population (children and adolescents)? , please explain in your manuscript.
3- Please define “ lethality”. Lethality means in-hospital mortality due to COVID-19? Or cause- specific mortality due to COVID-19 in community? Please define under study outcomes such as lethality and seroprevalence in your method .
4- In “ search strategies and screening” section, how many researchers have participated in search process? is the search process done independently? Please explain in your manuscript.
5- In “quality assessment” section, as mentioned “Two reviewers independently evaluated the quality of the studies “, if there was disagreement between reviewers about quality of an article, how did you resolve this disagreement? Please explain in your manuscript.
Results:
1- In “study characteristics” section that is mentioned “The sample sizes of the indigenous populations varied from 15 to 41,855”. I think this sentence is about sample of COVID-19 patients for reporting the lethality. Therefore, It is better to mention that this population is referred to COVID-19 cases.
2- Presentation of table 1 is ambiguous , it is recommended to;
• Present the information of seroprevalence and lethality in two separate tables.
• Present information of age and sex in separate columns.
• Please assign a number to each row (study).
• About seroprvalence and lethality rate, please add columns to separate numerator and denominator of population that studied. For example , total number of patients with positivity of serology test (numerator) present in a column, total number of patients that studied (denominator)present in another column and ratio of these values present in a column.

Annotated reviews are not available for download in order to protect the identity of reviewers who chose to remain anonymous.

Reviewer 2 ·

Basic reporting

The aim of the systematic review is to estimate the seroprevalence and lethality attributed to COVID-19 in indigenous populations of LAC. The chosen theme is current and relevant. The content presented is good, but it needs some repairs. Comments and suggestions below:
Introduction:
1. The text must be ordered, the information needs to be more interconnected. Example .
1st. Paragraph: write about SARS-COV2 infection, risk factors for its acquisition and progression to sever disease, explaining the social determinants that could increase exposure and more serious evolution. .
2nd. Paragraph: in addition to the cases described in LAC, tell about the social conditions that would facilitate the exposure and more serious evolution of this infection.
After this digression on the virus and LAC, describe the situation of indigenous people who should present high/greater exposure and worst evolution, justifying the objective of the review.
I believe that a reordering of the text will be enough.
2. English proofreading is required.

Experimental design

Aims
3. It is very clear and delimited. However, the estimation of prevalence and lethality was not made. They have been evaluated.

Methods
4. The research protocol was registered at Prospero.
Study eligibility criteria were well defined.
The search strategies in the various databases were adequately described.
The process of data collection and risk of bias assessment in the studies were satisfactorily described.
The authors planned to describe the data collected, without more complex analyses.

Validity of the findings

Results
5. Authors should add a flowchart of study selection, including the total for each database, number of articles excluded after reading titles and abstracts, etc... as recommended by Prisma (Moher et al. 2009). Please make two flowcharts if you feel it is necessary: for prevalence and lethality.
6. Authors must state the reason for exclusion from the analysis of those articles that were selected for full-text reading
7. Line 202: "Of the surveyed individuals, 66 and 36 were found to have SARS-CoV-2 antibodies after testing 203 with the WONDFO SARS-CoV-2 Antibody Test (21,25)." This excerpt can be deleted, as the next sentence speaks better (gives the %) of the same information

Discussion
8. Line 240-42 "Only a few non-241 representative studies were available that measured outcomes of COVID-19 in the indigenous people of LAC countries." I did not understand this sentence. Why non-representative?
9. L. 252 – 269. The authors discuss seroprevalence as if in your review the rate found was high and still citing studies carried out in LAC, but which were not included in the review, as an argument to affirm the vulnerability sounds indigenous in in relation to the general population. It would be better to discuss the possibility that the seroprevalence found in the review is underestimated.
10. L.271. Develop the discussion on COVID-19 lethality better, with special emphasis on the sample size of the studies and the lethality rate of hospitals in the region.
11. After paragraph line 306-320 in which you cite what other authors suggested in order to decrease morbidity/mortality of the indigenous people, it would be interesting to tell which measures that in your opinion should be taken, for the same purpose, such as a contribution to public health.
12. L.332-28. Elaborate better, as systematic review would not be possible without databases. Include in the limitations: 1. Few studies 2. Studies from few countries 3. Heterogeneous studies in relation to: - sample size in some studies, few included - diagnostic tests - form of recruitment and etc...
13. Conclusions L. 337-346 In this item of the manuscript you should only include the data of your review. The discussion and recommendations must be included in the Discussion item.

---

## Round 0.2 · accepted · Accept

Thank you for fully addressing the reviewers' comments. After reviewing Reviewer #2 concerns, it is my judgment that your manuscript in its current form is suitable for publication in PeerJ.

Reviewer 2 ·

Basic reporting

no comment

Experimental design

no comment

Validity of the findings

no comment

Additional comments

Dear editor,
The authors declined to make many of the suggested changes.
These suggestions were intended to include minimal scientific content, particularly in the background and discussion, as well as trying to correct some misconceptions.